# OpenReview forum: "Navigating the Safety Landscape: Measuring Risks in Finetuning Large Language Models"
_NeurIPS.cc/2024/Conference — NeurIPS 2024 poster_

### Official Review · Reviewer_jxYX · 2024-07-04

**Soundness:** 3
**Presentation:** 3
**Contribution:** 4
**Rating:** 7
**Confidence:** 3

**Summary:**

The paper introduces a novel concept called the “safety landscape,” which assesses the safety of generative language models. Within this landscape, the “safety basin” is defined as a safe local neighborhood around a model’s parameters. The key contribution is the introduction of a new metric called “Visage” that probes this safety landscape to determine how robust a model is against malicious fine-tuning. The authors present experimental results in both 1D and 2D safety landscapes, using various open-source language models, to demonstrate how Visage can help determine how robust models are against malicious finetuning and, in turn, help build more secure models.

**Strengths:**

The paper is well-written and presents a novel approach to model safety. The introduction of the safety landscape and safety basin concepts offers a novel perspective on evaluating model robustness. The introduced Visage metric is particularly valuable as it may aid in constructing models that are resistant to malicious fine-tuning in the future. This is especially significant for powerful open-source models, which are more vulnerable to such attacks. The experimental validation in both 1D and 2D safety landscapes provides a clear demonstration of the effectiveness of Visage.

**Weaknesses:**

- The paper uses the broad term “safety” to primarily describe a model’s refusal to answer potentially harmful queries. However, safety does not necessarily mean refusing to answer. For example, the authors use a refusal keyword detection mechanism to evaluate the safety landscape, but this approach has limitations. Safe responses can vary widely depending on the context, and simply refusing to answer can sometimes be considered unsafe (e.g. in the context of advice on self-harm). Further it does not cover context depending unsafe responses such as a response describing attributes of specific individuals or minority groups, which can be safe or unsafe depending on the content.

- Therefore the discussion on limitations could be more detailed.

Minor comments:
- Some related work is missing, such as https://openreview.net/pdf?id=6t0Kwf8-jrj, https://arxiv.org/pdf/2312.06681.
- Missing x-axis label in Figure 2 and labeling in Table 1 could be improved.

**Questions:**

- In Table 1, it is unclear what the “Aligned” column refers to. Can the authors clarify this?
- The paper's discussion of limitations is somewhat brief. Could the authors elaborate on the limitations of their current approach, including any potential weaknesses or challenges? This would also provide insights into potential future research to advance this interesting study.

**Limitations:**

The descriptions of limitations could be improved, see questions.

---

> ### Author Rebuttal · Authors · 2024-08-07
>
> We are grateful to the reviewer for acknowledging the significance of our findings and contributions! We hope the following clarifications can address the reviewer's concerns.
> 1. **Different refusal evaluation methods other than keyword search.**
> We agree with the reviewer that safety does not necessarily mean refusing to answer. Thus, we have expanded our experiments by testing an additional evaluation metric, Llama Guard 2. Our results demonstrate that the LLM safety basins exist regardless of the harmfulness evaluation metrics. Please check General Response -> Concern 1 for more details.
> 2. **Clarifications on x-axis label in Fig. 2 and Table 1.**
> The x-axis represents the scalar parameter $\alpha$ in Eq. 1, indicating the amount of perturbation added to the model’s original parameters. We have added annotations to the x-axis in all figures in the attached rebuttal PDF in General Response. We will also include these annotations in the revised version of the original paper. In Fig. 2a, the origin represents the Llama2-7B base model, and x-axis = 1 represents the Llama2-7B-chat model. In Fig. 2b, the origin represents the unperturbed model (Llama2-7B-chat), and all other points represent the measurement of ASR while perturbing the model weights along positive or negative directions. The “aligned” column in Table 1 refers to the original off-the-shelf models. We will clarify these annotations in the final version.
> 3. **More discussions on limitations and future work.**
> We believe there are multiple directions for future research, and our work is an important first step in exploring the safety landscape of popular open-source LLMs. Given our findings in the General Response, where we observe that the shape of the LLM capability landscape differs significantly from that of the LLM safety landscape, a potential direction for future work is to explore how to better balance the tradeoff between capability and safety, e.g., finding the optimal capability performance for a given dataset while staying within the safety basin. Another direction to explore, inspired by Reviewer h1px, is proposing additional sub-metrics such as basin width, depth, and smoothness. Our VISAGE score, defined as the average safety margin of all models sampled along random directions, can be considered as an average depth within the safety basin. The VISAGE score is a byproduct of our novel findings on the safety basin, and we hope our study will inspire further research into proposing more metrics, including width and smoothness.
>
> We sincerely thank the reviewer for all constructive feedbacks, and we will add the two related works in the “LLM safety alignment” of Sec. 2 in the final version.

---

> > ### Comment · Reviewer_jxYX · 2024-08-12
> >
> > Thank you for the additional experiments and clarifications. These have addressed my remaining concerns.
> >
> > I have no further questions.

---

> > > ### Author Response · Authors · 2024-08-12
> > >
> > > Thank you for your valuable comments and questions! We are grateful for your engagement in the rebuttal process. We are glad that we have addressed all your concerns! We will add those additional experiments and clarifications in the final version.

---

### Official Review · Reviewer_v82d · 2024-07-04

**Soundness:** 3
**Presentation:** 3
**Contribution:** 3
**Rating:** 8
**Confidence:** 5

**Summary:**

Inspired by the work of visualizing loss landscapes, the authors of this paper ask if there is a similar geometric interpretation of the weight space of LLMs and their respective vulnerability to answering unsafe questions. They provide a novel set of tools for perturbing the weight space of models, either along random directions or interpolated between two models. Based on these tools they suggest that there might be safety basins, a maximum radius after which weight perturbation quickly recovers unsafe behaviour and provide a measure of this radius called VISAGE. They observe that both model type and system prompts can have a large impact on the VISAGE score.

**Strengths:**

The notion of safety basins is a very interesting and novel phenomena which I think might have merit for understanding how to improve training-time and adversarial safety (make the basins larger!).

Regardless of whether safety basins actaully exist or not, the biggest strength of the paper is the development of tools for understanding weight perturbation and safety from a geometric perspective and I expect this to become a very important training dynamic analysis tool w.r.t to safety.

Weight perturbation as a defence against GCG and PAIR is also an interesting notion that they demonstrate empirically.

**Weaknesses:**

There is a conceptual clarity issue in the paper where loss landscape and “landscape drawn by ASR and weight perturbations” are being confused and muddled up in the motivations and through the paper when works drawing conclusions about the loss landscape are cited (IMO these should be removed as they are not relevant). I would advise the authors to thoroughly distinguish the two and make it clear to the reader how they are different and that your paper is not talking about loss landscapes. Along these lines “Model Landscape” is a very confusing and unclear term to me - Perhaps the clearest is attack success-weight landscape…

At this point, the paper has a critical experimental flaw: Since the ASR metric is a refusal keyword detector, there are many many alternative explorations for the “basin” shape the authors are getting through weight perturbation and no controlled study attempts to remove these confounders. For example, weight perturbation could just be generating gibberish text which would result in 100% ASR, or unrelated text, or text that is safe but otherwise doesnt’ use refusal keywords.

In order to recommend acceptance I need to see a few things:

(1) Measure a few (maybe 2-3) other capabilities like Mathematical reasoning for Random 1/2D perurbation - If there is a similar basin for all of these under perturbation then I think the paper will need to be rejected since we are just observing that perturbation ruins models which is obvious.
(2) Use a text fluency measure like perplexity in these perturbed regions.
(3) Use a few alternatives to ASR keyword measure (for all analysis such as for computing VISAGE) - In particular use a harmfulness measure like LLamaGuard which was designed for this purpose.
(4) Show a qualitative demonstrate of the text the model generates in each region (random selection of N samples - not cherry picked)

If the authors can provide these controls in the paper I would be willing to raise my scores since I would be convinced that safety basins do indeed exist and are an important phenomena to raise. I do acknowledge their observation on lines 187-188 but its an observation without experimental demonstration.

However there is another issue with the paper that would also limit me raising my score, safety is only evaluated using one dimension of safety: harmful question answering. I would encourage the authors to try to find safety basins in other cases of safety such as toxic content generation, phishing, bias,  cybersecurity and weapons development. Without this, I am concerned the safety basin finding would only be limited to harmful question answering.

**Questions:**

Can you clarify why Fig. 1 (B), section 4.3 is a novel finding? I might be missing something but since the hight of the basin is the ASR and harmful fine-tuning increases ASR, isn’t this what we would naturally expect? Is the innovation the plotting interpolated models on a direction to see when the ASR raises?  Or maybe its due to the radius of the allowed weight perturbations? Any clarity here would be appreciated.

Section 4.4: What were your findings on the size of the basin training with 100-shot safe alone? I think without this experimental result Table 1 is not properly controlled experimentally. (Ideally 100 random samples that are neither safe nor unsafe would provide an even better additional control!)

## Suggestions and Comments

2: adding “are” here is not grammatically correct

3: what does it refer to? safety alignment, not clear

13: The LLM Safety landscape

22: Did you mean to cite this paper for rejection sampling? or [25] instead?

79-80: I don’t think its correct to say advanced capabilites are attributed to safety alignment - I think the consensus is usually the opposite, that capabilities and aligment are largely orthogonal and that alignment imposes a “tax” (https://arxiv.org/abs/2112.00861) on capabilities.

86: Recent work has shown

106-107: Aren’t prompts a set of tokens and tokens what comprises prompts? I don’t think this distinction is clear to me. Perhaps Human prompt strategies versus optimization strategies is clearer.

127: suggestion - make it clear that i indexes each layer.

150: Since a recent user study

153: Tempature 0 is greedy decoding - https://arxiv.org/pdf/1811.02549

Figure 2  and others - Please label these access so its clear that these are the perturbation values.

1987-188: Provide this analysis in the appendix.

193-195: Provide this analysis in the appendix.

223: Can successfully measure

238-239: I don’t agree with this statement, the distribution of harmful samples used for evaluation and fine-tuning are very similar both drawn from the harmful question answering task. In order for it to be task-agnostic, you’d have to show the evaluation works across different types of unsafe distributions like toxic content generation, weapons development, bias&fairness.

277: System prompt design

Table 2: why were these not applicable? It would be nice to ihghlight the highest scores for clarity of reading.

289: space missing

325: safegaurds

326: are still effetive

**Limitations:**

I don’t think the authors provide adequate discussion of the limitations of either their perturbation tools, their VISAGE measure, or their experimental design. I have provided some suggestions above but some additional food for thought on the tools and measure are: What are the limitations of only selecting 1 or 2 dimensions? What geometry is being assumed for VISAGE? (i.e. can we find small perturbations within these norms that are sharp transitions to unsafe behaviour but everywhere else a wide flat safety basin?) Is that assumption justifiable?

---

> ### Author Rebuttal · Authors · 2024-08-07
>
> We are glad the reviewer finds our paper novel and considers the safety basin a very important tool for evaluating LLM safety finetuning. We also thank the reviewer for their constructive suggestions. We hope the following clarifications address the reviewer’s concerns:
> 1. **Measure a few other capabilities like mathematical reasoning for random perurbation.**
> We conducted additional experiments to evaluate on three datasets covering capabilities in math, history, and policy from MMLU. The shape of the LLM capability landscape is drastically different from the one in the LLM safety landscape; these landscapes do not exhibit the same trend, further highlighting our research’s novelty, and confirming the basin shape is indeed unique to the safety of LLM. Please check General Response -> Concern 2 for more details.
> 2. **Use a text fluency measure like perplexity in these perturbed regions and show qualitative demonstrations of the generated text.**
> We have conducted additional quantitative and qualitative experiments, which show that LLMs speak fluently even when ASR is high. We measured results quantitatively by using the perplexity on MTBench, and qualitatively by listing generated responses sampled along these directions. Please check General Response -> Concern 3 for more details.
> 3. **Different refusal evaluation methods other than keyword search.**
> We have expanded our experiments by testing an additional evaluation metric, Llama Guard 2. Our results demonstrate that the LLM safety basins exist regardless of the harmfulness evaluation metrics. Please check General Response -> Concern 1 for more details.
> 4. **Different safety datasets other than the AdvBench.**
> We have expanded our experiments by testing another safety dataset, policy-oriented safety evaluation (POSE) benchmark. This benchmark goes beyond harmful question-answering and includes unsafe instructions regarding hate/harness/violence, physical harm, malware, political campaigning, tailored financial advice, etc. Our results demonstrate that the LLM safety basins exist regardless of the and safety datasets. Please check General Response -> Concern 1 for more details.
> 5. **Remove the discussions on loss landscape in the motivation.**
> We want to emphasize that our paper strictly follows the definition of loss landscape in the original paper [1], defined as an empirical loss function (averaged over a set of data samples) of a neural network with low-dimensinal perturbations on the model weights for visualization. Our safety basin analysis falls within the loss landscape analysis by considering a binary 0-1 loss function defining the attack success of each attack query (safety keyword detection and Llama Guard 2). Following the reviewer’s comment, we will tighten the connection of our loss function to the loss landscape analysis in our revised version.
> 6. **In Table 2, why are certain system prompts not applicable to one LLM?**
> For Llama3, it’s because there is no default system prompt in the initial release, so we leave the default system prompt of Llama3 blank. For all other LLMs in the “safety” column, it’s because we are using the optimized safety prompts specific to each LLM from “On Prompt-Driven Safeguarding for Large Language Models” Appendix L. In the provided safety prompts, only Mistral overlaps with our research and there are no safety prompts provided for Llama2, Llama3, and Vicuna. We will make it clear in the paper why these are not applicable.
> 7. **Why Fig. 1B and section 4.3 is a novel finding?**
> Thanks for providing an opportunity for us to explain the novelty of our findings in these sections. The model’s performance **at the origin** is known from the literature, and we know for sure, that the model performance will degrade after adding malicious finetuning. However, how the model evolves from the origin to the breaking point is an important topic that is less studied in today’s LLM safety research. Is it a linear interpolation between the origin and the breaking point or is it maintaining the performance and making a sudden change? Our additional experiments in General Response -> Concern 2 exactly show that the shape of the LLM capability landscape is drastically different from the one in the LLM safety landscape; these landscapes do not exhibit the same trend, further highlighting our research’s novelty, confirming the basin shape is indeed unique to the safety of LLM. The gradual changes in the capability landscape align more with the common expectations, but the significantly different shape of the safety landscape is surprising and informative to future defenses!
> 8. **Additional results on finetuning with 100-shot safe dataset.**
> We provide additional results comparing finetuning with a 100-shot safe dataset to finetuning with a mixture of 100-shot unsafe and 100-shot safe datasets. The results show that finetuning with the safe dataset is more robust than finetuning with the mixture of both safe and unsafe datasets, and significantly more robust than malicious finetuning.
> | Model | VISAGE | AdvBench Samples | 100-shot unsafe | 100-shot safe + 100-shot unsafe | 100-shot safe |
> | :-- | --: |  --: |  --: | --: | --: |
> | Llama2-7B-chat | 85.32 | 520 | 95.4 | 0.2 | 0.1 |
> | Vicuna-7B-v1.5 | 73.26 | 520  | 96.7 | 1.2 | 1.0 |
>
> We sincerely thank the reviewers for thoroughly going through the paper and providing detailed and useful comments to help improve it! These editing suggestions and grammar issues will be addressed in the revised version.
>
> [1] Visualizing the Loss Landscape of Neural Nets

---

> > ### Comment · Reviewer_v82d · 2024-08-11
> > **Thank you for your great revisions**
> >
> > I want to thank the authors for their efforts for further revisions especially in a short time period the efforts are very much appreciated.
> >
> > I think that all of my concerns have been thoroughly answered and I have raised my scores accordingly. I also want to apologize for not considering 0-1 loss as a loss landscape measure, that was an oversight on my part.
> >
> > I want to emphasize that I think this is an important finding given the sharp drop-off of safety under perturbation versus capabilites. I would have raised my scores higher if there was a more comprehensive analysis of different capabilities under perturbation than MMLU and I could confirm that the paper has been revised to clearly state what is unique about safety, not that it is a basin - since capabilities "appear" to be basins as well, but that it is a basin with much sharper curvature. The sharp curvature providing excellent evidence that safety gaurding mechanisms appears to be much more brittle than capability degredation.

---

> > > ### Author Response · Authors · 2024-08-12
> > >
> > > Thank you so much for the positive feedback and for raising the score! We are glad that we have addressed all your concerns. We have also conducted additional experiments on the MT-Bench capability landscape. Following the official MT-Bench evaluation repo, we reported the capability scores evaluated by GPT-4, with scores on a scale of 10.
> > >
> > > Our findings show that the shape of the LLM capability landscape differs significantly from that of the LLM safety landscape (Fig. E in the code snippet), indicating that these landscapes do not exhibit the same trends. This distinction underscores the novelty of our research, confirming that the basin shape is indeed unique to LLM safety.
> > >
> > > Since we are not allowed to attach external links to figures during the discussion period, per the rebuttal instructions, we have included the following code snippet for plotting the MT-Bench capability landscape mentioned above. We will include a link to the figure if the AC permits.
> > > ```
> > > import matplotlib.pyplot as plt
> > >
> > > x = [-0.5, -0.45, -0.4, -0.35, -0.3, -0.25, -0.2, -0.15, -0.1, -0.05, 0, 0.05, 0.1, 0.15, 0.2, 0.25, 0.3, 0.35, 0.4, 0.45, 0.5]
> > > y = [0.25, 0.71, 1.57, 2.67, 2.57, 3.29, 4.88, 5.19, 5.73, 6.08, 6.23, 5.98, 5.58, 4.59, 3.81, 2.73, 2.42, 2.27, 1.21, 0.41, 0.32]
> > > plt.plot(x, y)
> > > plt.title('Fig. E: Llama2-7B-chat capability landscape on MT-bench')
> > > plt.xlabel("Perturbation Amount")
> > > plt.ylabel("GPT-4 score")
> > > plt.show()
> > > ```
> > > Thank you again, and we are glad we have addressed all your concerns!

---

> > > > ### Comment · Reviewer_v82d · 2024-08-12
> > > > **Thanks for sharing!**
> > > >
> > > > This certainly underscores the results for me!
> > > >
> > > > I do hope you add this to the work.
> > > > While I would normally not be confident in raising to an 8 without seeing the manuscript again, I will be raising my scores to an 8 due to the other reject score as I believe this also addresses their concerns.

---

> > > > > ### Author Response · Authors · 2024-08-12
> > > > >
> > > > > Thank you! Really appreciate your kindness! We will definitely add this.

---

### Official Review · Reviewer_KrSC · 2024-07-13

**Soundness:** 2
**Presentation:** 3
**Contribution:** 2
**Rating:** 4
**Confidence:** 5

**Summary:**

This paper aims to measure the LLM’s robustness against fine-tuning attacks by introducing the concept of “safety basin”. A new metric, VISAGE score, is proposed to measure the risk in fine-tuning without the need to actually fine-tune the LLM using a harmful dataset. The experiments demonstrate the proposed VISAGE score has a positive correlation with the robustness of LLMs against fine-tuning attacks.

**Strengths:**

1. This paper explains the success of fine-tuning attacks by navigating the LLM safety landscape.
2. The writing is clear and easy to follow.
3. The experiments demonstrate the proposed VISAGE score has a positive correlation with the robustness of LLMs against fine-tuning attacks.

**Weaknesses:**

1. The evaluation metric ASR used in Section 3 is not rigorous. Since ASR only captures refusal words, the increase in ASR may be because of the model’s decrease in utility, i.e., output random content after adding too much noise to the model weight. In this case, the ASR metric can also reach 100%. Therefore, I doubt the experimental results and the corresponding conclusions drawn in Section 3.
2. The conclusions drawn from the experiments in Sections 4 and 5 are not surprising. Specifically, the conclusions from Section 4.2 and Section 5 are already well-known [1][2]. In addition, since fine-tuning will compromise safety, it is natural that during the fine-tuning process, there is a gradual increase in ASR, which will form a “basin-like” shape in Figure 1 (b).
3. The experimental results are inadequate to support the claims. See Questions 1 and 3 for details.

[1] Fine-tuning Aligned Language Models Compromises Safety, Even When Users Do Not Intend To!

[2] Defending ChatGPT against Jailbreak Attack via Self-Reminder

**Questions:**

1. Is the basin in ASR really a safety basin? Or it is just the utility basin? Providing example outputs when the noise is large as well as the utility benchmark results (e.g., MT bench or Alpaca Eval 2, or just simply PPL) could be more convincing.
2. Why there is a safety basin for Vicuna? Vicuna doesn’t have a safety alignment.
3. Are there results of models other than Llama-2-7B-chat and Vicuna-7B-v1.5 in Table 1? Comparing Llama2 and Vicuna only for showing the high correlation between VISAGE and model safety might not be statistically significant.
4. What is the chat template for the Llama2-7B model? To my knowledge, there is no chat template for the base model, and the base model may not be used for chatting, except using URIAL.
5. Is there any comparison of costs between measuring model safety using VISAGE and performing fine-tuning attacks directly? Measuring the safety basin using VISAGE seems also computationally expensive.

**Limitations:**

Seems that the authors didn't mention limitations in the paper. There is no potential negative social impact of this work.

---

> ### Author Rebuttal · Authors · 2024-08-07
>
> We thank the reviewer for all the constructive suggestions, and we hope the following clarifications can address the reviewer's concerns:
> 1. **Does the model still generate fluent output when ASR is high?**
> We have conducted additional quantitative and qualitative experiments, which show that LLMs speak fluently even when ASR is high. We measured results quantitatively by using the perplexity on MTBench, and qualitatively by listing generated responses sampled along these directions. Please check General Response -> Concern 3 for more details.
> 2. **The conclusions drawn from Sections 4 and 5 are not surprising.**
> Thanks for providing an opportunity for us to explain the novelty of our findings in these sections. The model’s performance **at the origin** is known from the literature, and we know for sure, that the model performance will degrade after adding large perturbations at a certain point. However, how the model evolves from the origin to the breaking point is an important topic that is less studied in today’s LLM safety research. Is it a linear interpolation between the origin and the breaking point or is it maintaining the performance and making a sudden change? Our additional experiments in General Response -> Concern 2 exactly show that the shape of the LLM capability landscape is drastically different from the one in the LLM safety landscape; these landscapes do not exhibit the same trend, further highlighting our research’s novelty, confirming the basin shape is indeed unique to the safety of LLM. The gradual changes in the capability landscape align more with the common expectations, but the significantly different shape of the safety landscape is surprising and informative to future defenses!
> 3. **Safety basin for Vicuna.**
> Vicuna is finetunined from a Llama base model using user-shared conversations gathered from ShareGPT.com, a website where users can share their ChatGPT conversations. ShareGPT contains numerous portions of instruction tuning data related to safety. These data include supervised responsible answers like “I'm sorry, I cannot provide information on harmful or illegal activities” and “I am not able to provide personal opinions or evaluations of individuals.” Finetuning on such datasets is essentially the same as safety supervised finetuning (SFT), which is an essential step in most LLM’s safety alignment, e.g., Llama2 and Llama3. However, Vicuna is not further finetuned with DPO, RS, or other RL methods, thus making the model less robust than Llama models (lower VISAGE score), but it still possesses safety to a certain extent.
> 4. **Chat template of Llama2-7B base model.**
> The chat template in Fig. 2a in the paper is a general term defining the format of the conversation used in the evaluation. While interpolating the model weights between the base and the chat model, we need to ensure the chat format remains consistent. However, the base and the chat model use different chat formats, thus we ablate on both chat formats. Our results in Fig. 2a show that the chat model exhibits higher safety than the base model as expected. However, the model also shows a drastic increase in safety while using the Llama2-7b-chat chat template. We will add the above clarifications in the final verison.
> 5. **Computation time of the VISAGE score.**
> We use a single node A100 for safety landscape computation, and it takes ~12min to plot the landscape along a 1D random direction. Meanwhile, finetuning on 100-shot samples for 5 epochs takes ~16min under the same hardware configurations. Note that finetuning requires a model to finetune on a set of representative downstream tasks (significantly larger than 100-shot samples) and evaluate its safety, while our VISAGE definition does not make assumptions on the downstream finetuning dataset, serving as a task-agnostic safety metric that measures finetuning risks, which are also verified by our additional results on Llama Guard 2 and POSEBench in General Response. We also hope that our research can inspire future work that further accelerates the computation time of our new metric.
> 6. **Finetuning results other than Llama-2-7B-chat and Vicuna-7B-v1.5.**
> We finetune Llama3-8B-instruct with the default safety system prompt from Llama2. Our VISAGE score clearly indicates model’s safety after malicious finetuning.
> | Model | VISAGE | AdvBench Samples | Aligned | 10-shot | 50-shot | 100-shot |
> | :-- | --: |  --: |  --: | --: | --: | --: |
> | Llama3-8B-instruct | 90.40 | 80 | 0 | 87.5 | 90 | 98.8 |
> | Llama2-7B-chat | 85.32 | 80 | 0 | 90.0 | 91.3 | 100.0 |
> | Vicuna-7B-v1.5 | 73.26 | 80 | 5.0 | 95.0 | 97.5 | 100.0 |

---

> > ### Comment · Reviewer_KrSC · 2024-08-09
> > **Thank you for the detailed responses**
> >
> > Thank you for the detailed responses! Most of my concerns are well addressed. However, I still have two questions:
> > 1. Since you are using MT-bench questions, could you also report MT-bench scores so it can be more comprehensive?
> > 2. I still don't quite get the chat template you are using for the base model. Can you please specify it by giving the exact chat template you are using?
> >
> > Looking forward to your response!

---

> > > ### Author Response · Authors · 2024-08-12
> > >
> > > Thank you for your response! We are pleased to hear that we have addressed most of your concerns. Below, we provide clarifications to the two additional questions you raised:
> > >
> > > 1. We followed the official MT-Bench evaluation repo and reported the capability scores evaluated by GPT-4, with scores on a scale of 10. Our findings show that the shape of the LLM capability landscape differs significantly from that of the LLM safety landscape (Fig. E in the code snippet), indicating that these landscapes do not exhibit the same trends. This distinction underscores the novelty of our research, confirming that the basin shape is indeed unique to LLM safety. As we are not allowed to attach external links to figures during the discussion period per the rebuttal instructions, we have included the following code snippet for plotting the MT-Bench capability landscape mentioned above (we will include a link to the figure if AC permits):
> > > ```
> > > import matplotlib.pyplot as plt
> > >
> > >  x = [-0.5, -0.45, -0.4, -0.35, -0.3, -0.25, -0.2, -0.15, -0.1, -0.05, 0, 0.05, 0.1, 0.15, 0.2, 0.25, 0.3, 0.35, 0.4, 0.45, 0.5]
> > >  y = [0.25, 0.71, 1.57, 2.67, 2.57, 3.29, 4.88, 5.19, 5.73, 6.08, 6.23, 5.98, 5.58, 4.59, 3.81, 2.73, 2.42, 2.27, 1.21, 0.41, 0.32]
> > >  plt.plot(x, y)
> > >  plt.title('Fig. E: Llama2-7B-chat capability landscape on MT-bench')
> > >  plt.xlabel("Perturbation Amount")
> > >  plt.ylabel("GPT-4 score")
> > >  plt.show()
> > > ```
> > >
> > > 2. You are correct that the base model doesn’t use a chat template. In the paper, the term “template” refers to a preprocessing step applied to the raw user input to ensure it aligns with the model’s chat requirements. For the base model, this means there is literally no template; it is simply text completion. We will clarify this in the revised version.
> > >
> > > We hope our responses have fully addressed your concerns. We look forward to hearing from you and would be happy to address any remaining issues you may still have. If there are no further concerns, we kindly ask that you consider raising the score.

---

> > > > ### Comment · Reviewer_KrSC · 2024-08-13
> > > > **Thank you for your further clarification.**
> > > >
> > > > Thank you for your further justification and this is a genius way of putting figures :)
> > > >
> > > > From the figure, it seems that if the perturbation amount is lower than -0.25 or higher than +0.25, the utility is greatly degraded. Therefore, I still believe there is a correlation between the capability landscape and the safety landscape, and I suggest the authors discuss it in detail in the revised version of your paper.
> > > >
> > > > For my second question, I am still confused. If no template is provided, base LLMs will only do completion, i.e., continue writing the user query or instruction. Therefore, comparing the instruction model with the base one is meaningless.
> > > >
> > > > Overall, most of my concerns in my initial review are addressed and I thank you again for the hard work during the rebuttal. However, I believe there is still space to improve the quality of your work, as mentioned above.
> > > >
> > > > I will raise my score to 4.

---

> > > > > ### Author Response · Authors · 2024-08-13
> > > > >
> > > > > Thank you for your response.
> > > > >
> > > > > 1. We want to emphasize that the shape of the LLM capability landscape is drastically different from that of the LLM safety landscape. These landscapes do not exhibit the same trends, further highlighting the novelty of our research. This confirms that the basin shape is indeed unique to LLM safety. **During the discussion period, Reviewer v82d also acknowledged that our paper presents an important finding. The safety basin exhibits a much sharper curvature than the capability basin, providing strong evidence that safety drops faster under perturbations compared to capabilities.**
> > > > > 2. **Comparing the Base model with the Chat model is meaningful as the interpolation landscape reveals how safety changes between these two models.** The Chat model is known to be safer due to its training with multiple safety alignment methods, while the Base model is generally considered unsafe. To visualize the evolution of safety between the Base and Chat models, we need to account for the fact that both models use different “templates.” Fig. 2a confirms the safety of the Chat model and the unsafeness of the Base model, while also showing that the Base model is not completely compromised if the Chat model’s safety system prompt is applied.
> > > > >
> > > > > We hope our responses have fully addressed your concerns. We look forward to your feedback and will be sure to incorporate additional experiments and clarifications in the final version.

---

### Official Review · Reviewer_h1px · 2024-07-13

**Soundness:** 3
**Presentation:** 4
**Contribution:** 4
**Rating:** 8
**Confidence:** 3

**Summary:**

This paper looks at how robust/sensitive LLMs are in terms of safety training and finetuning. The authors study how robust models are by studying a "safety landscape" through perturbing the model's parameters in a random direction and evaluating the safety of the new perturbed model. They find many models exhibit a "safety" basin, or a region where the model is safe, with a sharp rise in unsafety after a certain point. The main contributions are:
- The authors propose a new metric VISAGE that studies safety by looking at the safety landscape
- They provide analysis across four open sourced models, showing a similar phenomena of a safety basin in all of them
- They further show VISAGE's usefulness by analyzing formulations like different system prompts and finetuning with unsafe data.

**Strengths:**

The paper has many strengths
- The paper is well-written and easy to read.
- The paper provides a novel method for analyzing how sensitive the safety training of an LLM is.
- The proposed metric is practically useful and can be generally applied to any LLM to investigate how secure safety training of the LLM is
- The authors provide comprehensive analysis of popular open sourced LLMs
- The authors provide practical insights using their metric, such as the effect of system prompts on safety as well as how finetuning effects the safety of LLMs and even a new direction to looking into of potentially thwarting jailbreaks.
- I can see this as a potentially useful metric to look at in the future when aligning LLMs: Wide safety basins are more preferable than narrow ones.

**Weaknesses:**

The main limitation of potential use of the method proposed in the paper is it seems this metric could be computationally intensive to compute, especially for really large models: Computing the VISAGE score requires approximating an expectation (Equation 5) over "average safety margin of all models we have sampled along all random directions" the authors mention they found that 3 random directions was enough, however it seems it is still necessary to sample many values of multipliers alpha and beta, applying the perturbation and evaluating the new perturbed model.

There is also a minor limitation of the refusal evaluation method used in all results (keyword search) not being the most accurate as the authors mentioned.

**Questions:**

I was curious about the following:
- VISAGE mainly looks at the average safety margin, I wonder if it could be possible/useful to divide this into a few more submetrics such as basin width (wider basins seem like they are more "robust), basin depth, and how smooth/bumpy the basin is.
- Could be interesting to see in a future work if there is any effect of Model size (ex 7B models vs 13B models)
- Could also be interesting in the future work to see if there is a similar basin for capability evals like MMLU
 - The authors mention on line 70 "A naive defense method [to jailbreaks] is to perturb the model weights before generating the response. ", it could be good to see how perturbations affect capabilities before consider a defense along these lines.

**Limitations:**

Generally I think this work shouldn't have any negative societal impacts - it could be possible that bad actors use the method to find effective jailbreaks on open-sourced models, but at that point, they could also through less effort just finetune the models to output harmful content.

---

> ### Author Rebuttal · Authors · 2024-08-07
>
> We are grateful to the reviewer for acknowledging the significance of our findings and contributions! We especially appreciate that the reviewers find our metric practically useful and can be applied to future LLM safety training analysis. We hope the following clarifications can address the reviewer's concerns.
>
> 1. **The effect of model size on safety basin (7B vs 13B models).**
> We expand our experiment by scaling up the model size from Llama2-7B-chat to Llama2-13B-chat. Fig. D (PDF in General Response) plots the 1D safety landscape of both models. Interestingly, a larger model size exhibits a wider safety basin, which also echoes the reviewer’s point of view that a wider basin seems to be more robust and a potential training goal for future LLM training.
> 2. **LLM’s capability landscape & How perturbations affect capabilities.**
> We conducted additional experiments to evaluate on three datasets covering capabilities in math, history, and policy from MMLU. The shape of the LLM capability landscape is drastically different from the one in the LLM safety landscape; these landscapes do not exhibit the same trend, further highlighting our research’s novelty, and confirming the basin shape is indeed unique to the safety of LLM. Please check General Response -> Concern 2 for more details.
> 3. **Different refusal evaluation methods other than keyword search.**
> We have expanded our experiments by testing an additional evaluation metric, Llama Guard 2. Our results demonstrate that the LLM safety basins exist regardless of the harmfulness evaluation metrics. Please check General Response -> Concern 1 for more details.
> 4. **Computation time of the VISAGE score.**
> We use a single node A100 for safety landscape computation, and it takes ~12min to plot the landscape along a 1D random direction. Meanwhile, finetuning on 100-shot samples for 5 epochs takes ~16min under the same hardware configurations. Note that finetuning requires a model to finetune on a set of representative downstream tasks and evaluate its safety, while our VISAGE definition does not make assumptions on the downstream finetuning dataset, serving as a task-agnostic safety metric that measures finetuning risks, which are also verified by our additional results on Llama Guard 2 and POSEBench in General Response -> Concern 1. We also hope that our research can inspire future work that further accelerates the computation time of our new metric.
> 5. **More sub-metrics such as basin width, depth, and smoothness in future work.**
> Thanks for the great suggestion! Our VISAGE score is defined as the average safety margin of all models sampled along all random directions, which can be thought of as the average depth in the safety basin. The VISAGE score is a byproduct of our novel findings on safety basin and we hope our study can inspire more future research on proposing more metrics including width and smoothness.

---

> > ### Comment · Reviewer_h1px · 2024-08-12
> >
> > Thank you for the response!

---

> > > ### Author Response · Authors · 2024-08-12
> > >
> > > Thank you for your valuable comments and questions! We are grateful for your engagement in the rebuttal process. We will add those additional experiments and clarifications in the final version.

---

### Author Rebuttal · Authors · 2024-08-07

# General Response
We sincerely thank all reviewers for their thoughtful feedback. We are excited that they highlight the strengths of our paper:
- **Safety basin, a new phenomenon observed universally in popular open-source LLMs, contributes significantly to the AI safety community and beyond.** Our notion of the “safety basin” is a very interesting and novel concept for evaluating LLM’s model robustness (h1px, v82d, jxYX). The comprehensive analysis of popular open-source LLMs (h1px) offers the AI safety community a new set of tools for visualizing the safety impact of model weight perturbations (v82d) and opens up future directions of how to improve LLM’s safety alignment by making the safety basin larger (h1px, v82d).
- **Practically useful and valuable new VISAGE metric for future LLM alignment research.** Our novel VISAGE metric probes into the safety basin and measures the safety of an LLM’s local region in model parameter spaces (jxYX, h1px). VISAGE shows a clear demonstration of its value in determining how robust a model is against malicious finetuning (jxYX, KrSC), which may become a very important training dynamic analysis tool w.r.t LLM safety (v82d). The metric is practically useful and can be generally applied to any LLM to investigate how secure the safety training is (h1px).
- **Our safety basin research provides new insights on the design of LLM system prompts and jailbreaking attacks and defenses.** The safety basin highlights the system prompt’s critical role in protecting a model, and that such protection transfers to its perturbed variants within the safety basin (h1px, v82d). In terms of jailbreaking attacks, we empirically show that weight perturbation may serve as a defense against GCG and PAIR attacks (h1px, v82d).
Finally, we appreciate that reviewers find our paper well-written and easy to follow (h1px, KrSC, jxYX).

Below we address several concerns that reviewers have shared:
### Concern 1: Does the finding of the safety basin generalize to other evaluation metrics and safety datasets? (v82d, KrSC)

**We have expanded our experiments based on the reviewers’ suggestions to test an additional evaluation metric, Llama Guard 2 [1], and another safety dataset, policy-oriented safety evaluation (POSE) benchmark [2]. Our results demonstrate that the LLM safety basins exist regardless of the harmfulness evaluation metrics and safety datasets.**

**Harmfulness evaluation metrics.** We replace the safety keyword detection with Llama Guard 2 to evaluate whether the generated output is safe or not. Llama Guard 2 is an 8B parameter Llama3-based LLM safeguard model. It classifies content as safe or unsafe, and if unsafe, it also lists the content categories violated. As shown in Fig. A, Llama Guard 2 evaluation also shows a basin shape similar to the safety keyword detection.

**Safety dataset.** POSE benchmark is constructed based on the exhaustive lists of 11 prohibited use cases found in Meta’s Llama-2 usage policy and OpenAI’s usage policy. We evaluate the generated outputs using both safety keyword detection and Llama Guard 2. Fig. B clearly shows that on the new dataset, both evaluation metrics show a similar basin shape.

### Concern 2: what does the capability landscape look like? Is it the same as the safety landscape? (h1px, KrSC, v82d)

**We conducted additional experiments to evaluate on three datasets covering capabilities in math, history, and policy from MMLU [3], as suggested by h1px and v82d. The shape of the LLM capability landscape is drastically different from the one in the LLM safety landscape; these landscapes do not exhibit the same trend, further highlighting our research’s novelty, confirming the basin shape is indeed unique to the safety of LLM.**

We evaluate capabilities using the following three datasets from MMLU: abstract_algebra, high_school_us_history, and us_foreign_policy datasets. Fig. C presents the results of perturbing the Llama2-7B-chat weights along a 1D random direction. For controlled comparisons, all datasets are evaluated along the same random direction. We observe that the shape of the capability score varies significantly across different datasets. For example, in the abstract_algebra dataset, the model also peaks at $\alpha$ (x-axis) = 0.2, while in the us_foreign_policy dataset, the model achieves slightly better performance at $\alpha$= 0.15. In contrast, randomly perturbing model weights maintains the safety level of the original aligned model in its local neighborhood, showing a rapid decrease in safety at the brim of the basin. Such drastic changes are not observed in the capability landscape. The gradual changes in the capability landscape align more with the common expectations, but the significantly different shape of the safety landscape is surprising!

### Concern 3: Does the model still generate fluent output when ASR is high?  (KrSC, v82d)

**We have conducted additional quantitative and qualitative experiments, which show that LLMs speak fluently even when ASR is high.** We measured results quantitatively by using the perplexity on MTBench, and qualitatively by listing generated responses sampled along these directions. We evaluate the perplexity of the perturbed Llama2-7B-chat model along a random direction using all 80 prompts from MTBench [4]. Fig. D demonstrates that the model maintains high fluency (low perplexity), even when ASR is high, except at the extremes (abs($\alpha$) > 0.4). Table A shows five responses sampled equally along the random direction ($\alpha$ = -0.5, -0.25, 0, 0.25, 0.5). At $\alpha$ = -0.25, we clearly observe the model speaks fluently but fails to refuse the harmful input.

[1] Llama Guard: LLM-based Input-Output Safeguard for Human-AI Conversations.
[2] Fine-tuning Aligned Language Models Compromises Safety, Even When Users Do Not Intend To!
[3] Measuring Massive Multitask Language Understanding.
[4] Judging LLM-as-a-Judge with MT-Bench and Chatbot Arena

---

### Decision · Program_Chairs · 2024-09-25

**Decision:**

Accept (poster)

**Comment:**

The paper aims to measure the risks of finetuning LLMs to compromise safety alignment. Specifically, the introduced VISAGE safety metric measures the safety of LLM finetuning by probing its safety landscape. AI safety is indeed an important problem (though we may not achieve the goal meaningfully, though the term suggests this), and the paper is well written. Still, the reviewers point out some drawbacks, such as being computationally intensive to calculate. Though true this might be addressed in future work. One of them even argues that there are no new results in terms of knowing which LLMs get broken. True, but the present paper addresses other aspects, such as a general metric, different rates, defending with benign examples, ..., and avoiding evaluating on some validating set (for the score). Indeed, the discussion about the correlation between the capability landscape and the safety landscape is still open, but it also indicates that VISAGE is interesting. So, while the authors have to add some discussion of downsides, there are enough pros to accept the paper. please note that the idea of computing a score and direction for safety has already been proposed in, e.g. (Schramwoski et al., Nat. Mach. Intell. 4(3): 258-268, 2022) for a single model. The present paper now uses this with a fine-tuning step in between.